# Password Cracking with Brute Force Algorithm and Dictionary Attack Using Parallel Programming

Ibrahim Alkhwaja [1], Mohammed Albugami [1], Ali Alkhwaja [1,*], Mohammed Alghamdi [1], Hussam Abahussain [1], Faisal Alfawaz [1], Abdullah Almurayh [2] and Nasro Min-Allah [1]

1   Department of Computer Science, College of Computer Science and Information Technology, Imam Abdulrahman Bin Faisal University, P.O. Box 1982, Dammam 31441, Saudi Arabia
2   Deanship of Admissions and Registration, Imam Abdulrahman Bin Faisal University, P.O. Box 1982, Dammam 31441, Saudi Arabia
*   Correspondence: 2190000164@iau.edu.sa; Tel.: +966-55-899-3488

**Abstract:** Studying password-cracking techniques is essential in the information security discipline as it highlights the vulnerability of weak passwords and the need for stronger security measures to protect sensitive information. While both methods aim to uncover passwords, both approach the task in different ways. A brute force algorithm generates all possible combinations of characters in a specified range and length, while the dictionary attack checks against a predefined word list. This study compares the efficiency of these methods using parallel versions of Python, C++, and Hashcat. The results show that the NVIDIA GeForce GTX 1050 Ti with CUDA is significantly faster than the Intel(R) HD Graphics 630 GPU for cracking passwords, with a speedup of 11.5× and 10.4× for passwords with and without special characters, respectively. Special characters increase password-cracking time, making the process more challenging. The results of our implementation indicate that parallel processing greatly improves the speed of password-cracking techniques. The brute force algorithm achieved a speedup of 1.9× with six cores, while the dictionary attack showed a speedup of 4.4× with eight-core static scheduling. Studying password-cracking techniques highlights the need for stronger security measures to protect sensitive information and the vulnerability of weak passwords.

**Keywords:** parallel computing; password cracking; brute force algorithm; dictionary attack; parallel programming; Hashcat; CUDA

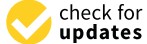



## 1. Introduction

In the domain of computer security, parallel computing has proven to be an effective method of accelerating the process of cracking passwords using brute force techniques as well as the dictionary attack. Brute force techniques involve trying every possible combination of characters until the correct password is obtained [1]. The duration of computation needed to discover a password by the brute force technique is reliant on various factors, including the length and complexity of the character set of the password, as well as the computational complexity of the encryption algorithm employed [2]. In addition, if the computation is carried out on a single processor, it may require even more time to crack the password. However, distributing the workload across multiple processors or devices can contribute to accelerating the process. This not only makes password cracking more efficient, but also increases the chances of having a successful cracking attempt. When a password is deemed to be highly secure, brute force password cracking becomes indispensable, but there are alternate methods for cracking passwords that are considered more favorable, such as the dictionary attack [3]. The dictionary attack uses a pre-compiled list or wordlist of commonly used passwords and matches them with the targeted password [4]. This method is less time-consuming compared to brute force but is only effective if the password exists in the list used. Therefore, it is usually used when the brute force attack takes too long to crack lengthy passwords [5]. Again, parallel

programming can be utilized to reduce the time it takes to perform the attack. Nevertheless, parallel computing can also be used to implement highly optimized password-cracking algorithms, such as using GPUs for processing large amounts of data. Can et al. [6] argue that the difficulty of the algorithm determines how much computation is required. A study conducted by Laatansa et al. [7] investigated the effectiveness of cracking SHA-1 hashed passwords using a GPGPU-based machine with brute force and dictionary attack methods. The results demonstrate that brute force is more effective in cracking passwords with fewer characters, with 11% of passwords containing seven or fewer characters cracked, compared to only 3% with the dictionary attack. On the other hand, dictionary attack proves to be more effective in cracking passwords with insecure character patterns, with 5053 passwords cracked versus 491 with the best brute force scenario. A combination of both methods (brute force and dictionary) provides a more balanced approach to cracking passwords, regardless of length or character pattern security [7]. The use of parallel computing in password cracking is essential not only for security professionals to empower their defense systems by identifying weak passwords used by employees, for instance, but also for individuals and organizations who may have lost or forgotten their passwords. Most operating systems and apps use key derivation functions (KDFs) to transform plaintext passwords to hashed passwords in order to prevent attackers from simply obtaining the clear text password. A brute force attack would be the only method to recover the plaintext password from a hashed password since KDFs are one-way functions [8]. Using parallel computing in password recovery is essential as it saves time and retrieves data and information protected by a forgotten password. On average, 76% of internet users use the same password across other websites, according to [9]. This increases the chance of their accounts being compromised. Another technique of cracking passwords is the dictionary attack. A predetermined list of common passwords is used in the dictionary attack to achieve a potential match. This technique poses a significant threat to various entities, particularly in the realm of account and network security. For instance, it can involve exploiting Wi-Fi networks by targeting common passwords to gain unauthorized access [10]. The objective of this study is to determine the most efficient and effective approach for password cracking by evaluating the performance of various hardware configurations for parallel brute force and dictionary attacks. Furthermore, the objective aims to investigate the impact of character sets on password-cracking performance, specifically the inclusion of special characters. The results of this study demonstrate the significant impact that hardware configurations, such as multiple cores or powerful GPUs, can have on the efficiency of password-cracking techniques. By leveraging these advanced hardware tools, security professionals can achieve a dramatic increase in password-cracking speed and accuracy, which is critical in the fight against cyber threats. These findings underscore the importance of ongoing research and development in the field of hardware configurations for password cracking and suggest that future advances in this area could have a transformative effect on the field of cybersecurity.

The rest of this work is structured in the following manner. In Section 2, a review of 16 related papers of literature is presented. The proposed techniques are given in Section 3. The proposed techniques used are as follows: implementing parallel processing in brute force and dictionary attacks using Python language, implementing the dictionary attack with OpenMP in C++, and implementing brute force and dictionary attacks using Hashcat. In Section 4, there are research studies that encompass both the experimental design and the optimization technique or method used to search for parameters. In Section 5, results and discussion can be found. The analysis and comparison with other research studies are presented in Section 6. Finally, Section 7 contains the conclusion and suggestions derived from the implementation.

## 2. Literature Review

Ignatius and Yusuf [11] discussed the use of the CUDA computing platform to support a brute force algorithm as it requires a large number of computational resources. The

authors evaluated five factors that may affect the performance of a GPU-based parallel program. The elements considered were integer arithmetic, delay in accessing memory, data transfer, shared memory usage, and utilization of registers. They created custom and test algorithms to assess these factors based on prior research on cracking PDF passwords. The final algorithm was constructed by incorporating the factors that had the greatest impact. The parallel algorithms were put into practice on a Tesla C2075, and the results showed a speedup of 2.92 for 2-byte alphanumeric passwords and 4.77 for 6-byte numeric passwords. The study included five factors in both the test and testbed algorithms, and the results showed that shared memory has the greatest impact on the algorithm's performance. The shared memory factor helped the algorithm to achieve a speedup of up to 4.77.

Sarah and Robert, in their study [12], investigated the potential of using low-power and energy-efficient hardware for password cracking. They argued that this type of hardware could be useful in security applications that do not require high-speed processing, such as long-term surveillance or battlefield scenarios. The study described a proof-of-concept implementation using the Epiphany series of chips from Adapteva Inc., Lexington, MA, USA, which have a theoretical peak performance of 16 GFLOPS/W and a power consumption limit of 2 W. The experiments were performed using a symbol set of size 10, generating a total of 100 plain texts of length 4, which were encrypted using the SHA-512 algorithm. The study compares the performance of a brute force algorithm that is executed sequentially on a dual-core ARM Cortex-A9 host, as well as a parallel implementation designed for the Epiphany co-processor. The results indicated that the parallel implementation provides an energy-efficient means of executing the algorithm, with a speedup of up to 16 times that of the serial version. The authors also mentioned the limitations of the implementation, such as the limited amount of available RAM and the need to consider the memory bottleneck. Overall, this study provided a proof of concept for energy-efficient brute force password cracking using low-power hardware and opens the door for further research in this area.

Can et al. [6] suggested an enhanced brute force password recovery method for SHA-512 on a GPU utilizing a variety of optimization techniques. To fully utilize the GPU, they used OpenCL, and they used C to develop programs that ran on the GPU. The Secure Hash Method (SHA-512), a one-way hash algorithm that is frequently used for password encryption and verification, is what they aimed to utilize in a brute force attack. They created a simple GPU system in their work and improved it using several optimization techniques, including password concatenation, register reuse, faster instructions, and a meet-in-the-middle strategy. The first section includes a basic password recovery scheme for SHA-512. In this scheme, the CPU generates passwords in batches, while GPUs are utilized to perform concurrent SHA-512 hash calculations. The next step is to implement optimizations at the program level, such as utilizing registers repeatedly and replacing certain instructions. In the third part, they apply additional algorithms to the basic scheme to make it more optimized. After all of their testing and research, they were able to achieve a speed of 1055 M hashes per second on two AMD R9 290 GPUs. Their solution is 11% quicker when compared to Hashcat, the quickest password-cracking program.

Feng et al. [13] discussed that UNIX systems frequently employ the cryptographic technique MD5 Crypt for authentication. It renders traditional password cracking approaches on widely used computer platforms and ensures system security by leveraging more salt randomization and greater scheme complexity. However, a brute force attack threatens MD5 Crypt security once again due to the rapid growth of petaflop heterogeneous supercomputer systems, such as Tianhe-1A. For MD5 Crypt's speed to increase, much work has been completed on the GPU-accelerated platform. Utilizing the CUDA architecture's constant memory, however, has not led to much of an improvement. Their research investigates the issue and reports a 44.6% improvement by giving the padding array constant memory. In addition, using Tianhe-1A, the world's fastest heterogeneous supercomputer, their study provides a highly scalable implementation of the MD5 Crypt Brute Force Attack Algorithm. According to their testing findings, a single processing node

can check 326,000 MD5 hashes per second, outperforming the CPU version by 5.7 times. Their solution also exhibits excellent scalability on multiple nodes. As a result, it presented a fresh problem for MD5 Crypt's authentication security.

In [14], Abdelrahman et al. explored the implementation of WPA-WPA2 PSK cracking on parallel systems. For making use of the parallel platforms, they ran their single-threaded cracking program on several systems. They also employed the dictionary attack method, which depends on how quickly the most pre-shared keys are tried. The shared memory concept was also implemented using Pthread, OpenMP on multi-core CPUs, and CUDA on the GPU. They developed a single-thread cracking program based on two files: a password dictionary file and a completed four-way handshake CAP file. The password dictionary file is used to retrieve the password, while the CAP file is used to extract the input parameters required for the cracking. They implemented the cracking tool on the GPU using CUDA, a C extension that allows programmers to utilize the processing capability of both the CPU and GPU. The results demonstrate that employing multi-core processors and a GPU on regular machines with parallel platforms increases cracking performance by $16\times$ and $41\times$, respectively. Similarly, ref. [15] proposed a new method to crack Wi-Fi passwords using GPU and parallel computing. The current WPA/WPA2 protocol is more secure but is still vulnerable to brute force cracking attacks. The proposed algorithm is effective and enhances the efficiency of cracking Wi-Fi passwords by restricting the combination range of dictionary files and using GPU and CPU for simultaneous parallel division of tasks. The speed of dictionary cracking using a combination of CPU and GPU processing was much faster than traditional CPU computing power. However, this method is useless for long passwords.

David et al. [16] experimented with how hashing can be sped up in hash function operations and string comparisons which can be time-consuming. However, these operations can easily be parallelized since each password can be tested separately. For the purpose of making the performance faster, high-performance computing (HPC) can be obtained. GPU computing can enhance performance even more. Multiple GPUs can be used for an even larger scale, but this increases communication latency, reducing overall performance. In this work, MPI was used to reduce communication latency and process data communication among machines. The paper demonstrates three password recovery algorithms that use both MPI and CUDA. The algorithms differed in GPU memory utilization and data distribution. The algorithms that involve dividing dictionaries and password databases showed great performance with a speedup of $17\times$ and $12\times$ using eight GPUs across four nodes, respectively. The minimal memory algorithm showed slower performance due to communication latency. The algorithms scale well to multiple GPUs and can be used for larger databases. This work can improve computer system security by identifying weak passwords and could be useful for processing huge amounts of data using MPI and CUDA.

Anh-Duy et al. [17] proposed a homogeneous parallel brute force algorithm that utilizes the GPU to crack passwords. They used the CUDA framework 3.0 to crack SHA1. In addition, they divided the passwords into five groups. The first group contains only numeric characters, the second group contains lowercase characters, the third group contains numeric and lowercase characters, the fourth group contains lower and uppercase characters (sensitive alphabet characters), and the last group contains numeric and case-sensitive characters. They measured the password cracking time for each group. The password contains six characters hashed using SHA1. Moreover, the proposed algorithm has a memory-consuming disadvantage. The test was performed on the Tesla C1060 with 240 cores and 1.3 GHz core clock speed and the Tesla C2050 with 480 cores and 1.15 GHz core clock speed. They were able to crack a six-character password in less than 1 s using the Tesla C2050.

Maruthi et al. [18] discussed that although the MD5 hash algorithm's FPGA implementation is quicker than its software equivalent, theoretically, 2128 iterations are required for a pre-image brute force attack on an MD5 hash. Their research makes an effort to accelerate hardware-implemented brute force attacks against the MD5 algorithm. For the generation

of MD5 hashes, a complete 64-stage pipelining is used, and for the generation of a guessed password, three architectures are given. The MD5 hash generator and password generator pair are parallelized using 32/34/26 instances to find a hashed password with the use of the MD5 technique. A single Virtex-7 FPGA chap was used to attain a total of around 60 trials/s performance.

In [7], Laatansa et al. argued that a system's password data are often hashed. Tzhe data may fall into the hands of people who are not authorized to access them or even those who have malevolent intentions due to a variety of human errors and system vulnerabilities. Brute force and dictionary attacks are examples of attacks that might be made on hashed password data using a GPGPU-based system. The researchers' study describes the efficacy of brute force and dictionary attacks carried out using GPGPU-based machines on SHA-1 hashed passwords. The findings demonstrate that password cracking using brute force methods is more successful on passwords with shorter lengths (over 11% of them had seven characters or less) than dictionary attacks, which only cracked 3% of them. When compared to the best brute force attack scenario, which takes 491 passwords, a dictionary attack is more effective in breaking passwords with unsafe character patterns (5053 passwords). A more balanced technique for password cracking, regardless of how lengthy or secure a password is, is provided by the use of a combined attack strategy (brute force + dictionary).

In [19], Qingbing and Hao published a paper about password recovery for WinRAR3 encrypted files without a file name encryption. The present cracking methods, which employ a single CPU or GPU platform, are constrained by the CPU's and GPU's sluggish decryption and decompression algorithms, respectively. This problem is addressed by the paper's efficient cracking approach, which uses CPU + GPU pipeline collaboration to speed up computation times and boost performance. In order to simplify the decompression computation, the approach additionally takes advantage of magic number matching. The testing findings demonstrate that the suggested strategy improves eight-digit password speed by 2.3 times. The study emphasizes the merits of WinRAR3's compression method and the significance of heterogeneous multi-core architecture for password cracking. The following phase in improvement entails further improving CPU and GPU collaboration as well as GPU and CPU performance. In a similar study to improve password recovery for archived files, ref. [20] suggests a new method for creating effective hardware accelerators for password recovery on FPGA devices using the RAR3 algorithm. This approach involves making adjustments to the data paths at two levels of granularity: coarse and fine. Coarse-grained adjustments are used to remove the randomness of message format resulting from variations in password length, while fine-grained adjustments meet the data requirements of the pipelined hashing unit when the password length is constant. This strategy enables efficient data scheduling via regular data interconnect paths, which improves efficiency and reduces overhead for password recovery accelerator components.

Guang et al. [21] published a study on how to perform password recovery for RAR files using CUDA. With an emphasis on the process of generating keys for AES encryption, the step that takes the most time in the RAR encryption/decryption process, the article describes research on password recovery for encrypted RAR files. A version that utilizes a CPU is also provided for comparison. The implementation is based on NVIDIA's CUDA. The study finds that the GPU performs better than the CPU but suggests that the SHA-1 algorithm may still need to be improved. Also suggested is a modified approach for calculating program performance.

Jaewoo et al. [22] assessed the security of pattern-based passwords used in Android devices by exhaustively searching for secret patterns with the help of GPU parallelism. The Android system employs a $3 \times 3$ grid for pattern locking, with a total of 389,112 permitted patterns that must adhere to certain restrictions, such as having a minimum of four points and avoiding intermediate points. The researchers aimed to determine the vulnerability of these passwords to hacking techniques such as shoulder surfing or smudge attacks by employing a brute force search method, which involves graph searches with four to

nine vertices and a maximum of eight edges. For the purpose of improving efficiency, the GPU-implemented brute force search algorithm is optimized and recursive. The algorithm explores different paths in the graph with multiple threads, each having a unique identifier to track the path, and octal representation is used for easier conversion. The study provides some initial results indicating the potential vulnerability of pattern-based passwords on Android devices.

The study presented in [9] focuses on recovering encrypted document passwords, which is in demand for forensic purposes. On standard hardware with multi-core CPUs or GPUs, the authors conduct an experimental study of the password recovery procedure. The findings demonstrate that recovery time may be estimated given the length, alphabet, and hardware performance of the password. The fundamental technique for recovering a password entails a thorough search (brute force). However, with the help of powerful GPUs, the operation can be finished faster than it would with a CPU-only computer. In addition, authors in [9] improved their program by adding more advanced password generators and compatibility for additional file types. They also concentrated on password recovery in distributed environments. In future studies, they intend to evaluate their technology against existing programs such as the AccessData Password Recovery Toolkit, version 7.6.0.

Similarly, another research paper [23] presents a solution for optimizing the password recovery process for RAR files, which are compressed archives that employ the SHA-1 hashing and the AES encryption algorithm. The authors use a coarse granularity parallel technique and concentrate on using GPUs to speed up password recovery. The strategy employs four optimization techniques to boost performance, including asynchronous parallel between CPU and GPU, redundant calculations and conditional statements reduction, data locality using LDS, and the usage of register optimization. According to the experiments, the optimized parallel version outperformed a well-optimized serial version on a CPU by a factor of 43 to 57. When using additional GPUs, the results likewise demonstrated linear performance acceleration. Their future work will concentrate on fixing the algorithm's remaining flaws, such as poor portability and a lack of password search improvements.

Finally, Zhendong and Peng [8] suggested a new accelerator design for sha256crypt password recovery that employs hybrid CPU-FPGA devices. The design was tested on the Xilinx Zynq 7000 XC7Z030-3 SoC, and the experimental results demonstrate that the proposed accelerator significantly improves energy efficiency, achieving a $2.54\times$ improvement compared to Hashcat running on an NVIDIA GTX1080Ti GPU. The accelerator also improves energy and resource efficiency by $1.64\times$ and $1.69\times$, respectively, compared to the pure FPGA-based implementation in John-the-Ripper. Additionally, the paper outlines the difficulties in using pipeline techniques and presents various strategies to reduce hardware resource overhead. The techniques can be applied to other key derivation functions (KDFs) with comparable features. The researchers also plan to address structural weaknesses and optimize the system using partial reconfiguration technology in the future.

Table 1 provides a summary of the related work, including the dataset used, the number of samples included, the number of threads or processors employed, the techniques utilized, and the best results achieved.

**Table 1.** Summary table of related work based on different criteria.

| Ref. | Dataset | Number of Samples | Number of Threads/Processors | Techniques | Best Result |
|---|---|---|---|---|---|
| [11] | 3906 for modulo and 4160 for bitwise. | None. | 32, 64, 128, and 256 threads per block. | Using CUDA with GPU for parallel password cracking and optimized shared memory for 6-byte numeric passwords. | 4.77 speedup. |

**Table 1.** *Cont.*

| Ref. | Dataset | Number of Samples | Number of Threads/Processors | Techniques | Best Result |
|---|---|---|---|---|---|
| [12] | None. | 100 plain texts of length 4. | 16 processors/threads. | Parallel implementation of a brute force algorithm for the Epiphany co-processor. | Speedup of up to 16 times that of the serial version. |
| [6] | None. | None. | 2560. | They used several optimization techniques: combination of passwords, repetition of register utilization, faster instructions execution, and meet-in-middle. | 1055 M hash/s. |
| [13] | None. | None. | Each node of Tianhe-1A supercomputer has two CPUs, one GPU. The CPU has 6 cores while the GPU has 448 cores operating. | Using both the CPUs and the GPU on one single node. | 326,000 MD5 hashes are searched per second, which is 5.6 times faster than the performance of the CPU-only version. |
| [14] | None. | 2. | The computer has an Intel i7-4710HQ processor (2.50 GHz, 4 physical cores, 8 threads). The multi-core version was tested on a different computer (Intel i7-2630 QM, 2 GHz, 4 cores, 8 threads) using Ubuntu 12.04. The GPU used was a GeForce GTX 860 M. | They used the shared memory model. | They archive the best score by using GPU platform on the Windows operating system with 1,000,000 passwords in 384 s. |
| [15] | None. | N/A. | The server has two 16-core CPUs, 192 GB memory, and four GeForce GTX 1080 graphics chips, each with 8 GB video memory. | Hybrid parallel processing using Multi-CPU-GPU for calculations. | The performance of a single-core GPU is about 80 times higher than that of a single-core CPU. |
| [16] | 1 M. | N/A. | 8 GPUs. | Parallel brute force using the GPU with MPI and CUDA. | $17\times$ speedup. |
| [17] | None. | 5. | The Tesla C1060 has 240 cores with 1.3 GHz clock speed; the Tesla C2050 has 480 cores and runs at 1.15 GHz. | Parallel brute force using the GPU. | They could crack a 6-character password in less than a second. |
| [20] | None. | None. | NVIDIA GTX 1060 GPU. | Dual-granularity data path adjustment strategy. | It achieved an improvement of 3.3 speedup in energy efficiency. |
| [18] | None. | There were three sample passwords to be cracked. | The FPGA they utilized, the Virtex-7, contains 485,760 logic cells and 75,900 slices. Four LUTs and eight flip-flops are found on each slice of the Xilinx 7 series FPGA. | Cracking a hashed password using their architecture-3 (Arch-3) with 26 instances, and it deals with passwords of alphabets only on Virtex-7 FPGA. | They were able to crack a hashed password with 7 digits within 156 s. |

**Table 1.** *Cont.*

| Ref. | Dataset | Number of Samples | Number of Threads/Processors | Techniques | Best Result |
|---|---|---|---|---|---|
| [7] | 6,458,020 hashed passwords. | None. | The used device's GPU is 2 × MSI GeForce RTX 2080 Ti Gaming X Trio. | Using brute force and dictionary attacks by utilizing the software Hashcat v5.1.0. However, CPU resources were not utilized at all; instead, tests were conducted using both RTX 2080 Ti units with CUDA with a limit of 30 min. | Brute force cracked 770,884 short passwords (6 to 7 characters) while dictionary attack succeeded in cracking 63,119 long passwords (9 or more characters). |
| [19] | None. | None. | CPU: Xeon(r)e5-2620 Number of cores: 8 Number of threads: 16. GPU: GTX 1080 Ti: 3584 Cores. | It uses CPU + GPU pipeline collaboration to speed up computation times and boost performance. In order to simplify the decompression computation, the approach additionally takes advantage of magic number matching. | The cracking algorithm now runs at 24,423 passwords per second, which is a 2.3 times better performance than its previous speed. |
| [21] | None. | None. | CPU: Intel Core2 Duo E7300: 2-core GPU: GTX 9800: 128 Cores. | They concentrated on the process of generating keys for AES encryption, which is the step that takes the longest during the RAR encryption/decryption process, and they used GPU instead of CPU because GPU performs better than CPU in this case. | The average time for computing a single key reaches a maximum when computing around 16,384 keys. The average time per key is estimated to be 1.5 ms. |
| [22] | None. | None. | Intel Xeon E5-2630 and Nvidia Quadro 4000 GPU with 256 CUDA cores. | Utilizing parallel processing on general-purpose GPU to conduct exhaustive graph search. | The study shows the limitations of Android's pattern lock system and establishes the foundation for developing tools that can assess the security of passwords based on patterns. |
| [9] | MySpace: 37,000. phpbb: 180,000. RockYou: 14,000,000. singles: 12,000. Facebook: 2441. pwgenl (generated): 1,000,000. | None. | GPU: features 2816 shaders and 64 ROPs. CPU: 4 cores + 4 with hyperthreading. | Exhaustive search. | 12% speedup. |

**Table 1.** *Cont.*

| Ref. | Dataset | Number of Samples | Number of Threads/Processors | Techniques | Best Result |
|------|---------|-------------------|------------------------------|------------|-------------|
| [23] | None. | None. | GPU: 1792 stream processors, 32 color ROPs. CPU: 4 cores + 4 with hyperthreading. | Asynchronous parallel between CPU and GPU. | The final version resulted in an increase in performance speed that ranges from 43 to 57 times. |
| [8] | None. | None. | Hybrid CPU-FPGA: Xilinx Zynq-7000 XC7Z030-3 SoC. | A Hybrid-CPU-FPGA-based solution. | Energy efficiency 2.54× compared with NVIDIA GTX1080Ti GPU. Energy and resource efficiency 1.64× and 1.69× compared to the pure FPGA-based. |

## 3. Proposed Techniques

In this experiment, we used four various techniques for the purpose of implementing parallel execution on the brute force attack and dictionary attack. All techniques were used to perform sequential and parallel tests to compare results. In addition, brute force and dictionary methods will be compared as they both were performed in Python with the same hardware characteristics.

### 3.1. Applying Brute Force Using "ProcessPoolExecutor" in Python #1

The ProcessPoolExecutor class, which is based on the concurent.future model, is imported to construct parallel tasks [24]. It provides the ability to run multiple functions in parallel by utilizing a pool of processes. The class splits tasks among several processes running on different cores of the computer, allowing functions to run in parallel. By using "ProcessPoolExecutor", you can easily submit functions to be executed simultaneously and retrieve their results when they are done. It also takes care of managing the process pool and distributing tasks to the processes, freeing you up to concentrate on writing your application logic. It also manages communication between the main process and its child processes. Figure 1 demonstrates how it can be imported.

```python
from concurrent.futures import ProcessPoolExecutor
```

**Figure 1.** Importing the class ProcessPoolExecutor from the concurrent.futures library in Python.

Figure 2 shows the code portion of brute force implementation using the "Proces.sPoolExecutor" class.

### 3.2. Applying Dictionary Attack Using "Multiprocessing" Module in Python #2

Using the multiprocessing package in Python allows multiple processors within a single computer to reach their full potential [25]. Multiprocessing in Python refers to the ability to run multiple processes simultaneously within a single program. The multiprocessing module provides a way to create separate processes for different tasks and communicate between them to coordinate the execution of the program. This allows for parallel processing, which can be useful for taking advantage of multiple cores and increasing the speed of computationally intensive operations. Figure 3 demonstrates how the module can be imported in Python.

```python
def attempt_match(length , target):
    for attempt in itertools.product("abcdefghijklmnopqrstuvwxyz0123456789", repeat=length):
        attempt = "".join(attempt)
        if attempt == target:
            print("Password was Found :", attempt)
            return attempt

if __name__ == '__main__':
    target = input("Enter the password to be cracked: ")
    Numb_of_cores = input("Enter the number of CPU core you want to use : ")
    start_time = time.perf_counter()

    with ProcessPoolExecutor(max_workers=int(Numb_of_cores)) as executor:
        results = [executor.submit(attempt_match, length , target) for length in range(1, len(target)+1)]

        for future in concurrent.futures.as_completed(results):
            match = future.result()
            if match:
                break
```

**Figure 2.** Using the class "ProcessPoolExecutor" to implement parallel processing.

```python
import multiprocessing as mp
```

**Figure 3.** Importing the multiprocessing library in Python.

Figure 4 shows a code portion to demonstrate the use of the module in a dictionary attack using Python.

```python
num_processes = 8
chunk_size = len(dictionary) // num_processes
processes = []
for i in range(num_processes):
    start_index = i * chunk_size
    end_index = (i + 1) * chunk_size
    p = mp.Process(target=password_check, args=(dictionary, target, start_index, end_index))
    processes.append(p)
    p.start()

for p in processes:
    p.join()
```

**Figure 4.** A segment of code using a multiprocessing library.

### 3.3. Applying Dictionary Attack Using OpenMP in C++ Technique #3

A set of environment variables, library functions, and compiler directives called "OpenMP" make it simple to parallelize sequential source code [26]. OpenMP allows us to add parallelism to dictionary attacks by adding a few lines of code without having to worry about the details of thread creation, synchronization, and communication. For this technique, we wrote a program in C++ to implement six different processing settings on a list of 14,442,064 unique passwords from "RockYou" available at Kaggle and compared the results to analyze them. The techniques implemented are as follows:

1.  Sequential code. Figure 5 illustrates the sequential code.

```cpp
for (int i = 0; i < dictionary.size(); i++) {
    if (target == dictionary[i]){
        cout<<"password found: "<< dictionary[i]<<endl;
        cout<<"number of i is: "<< i<<endl;
        cout<<"Number of thread is: "<< omp_get_thread_num()<<endl;
        cout<<"Total number of passwords is: "<<dictionary.size()<<endl;
    }
}
```

**Figure 5.** A segment of the sequential code.

2.  Parallel code with 4 cores using static scheduling. Figure 6 demonstrates the parallel code with 4 cores using static scheduling.

```cpp
#pragma omp parallel for num_threads(4) schedule(static)
for (int i = 0; i < dictionary.size(); i++) {
    if (target == dictionary[i]){
        cout<<"password found: "<< dictionary[i]<<endl;
        cout<<"number of i is: "<< i<<endl;
        cout<<"Number of thread is: "<< omp_get_thread_num()<<endl;
        cout<<"Total number of passwords is: "<<dictionary.size()<<endl;
    }
}
```

**Figure 6.** A segment of parallel code with 4 cores using static scheduling.

3.  Parallel code with 8 cores using static scheduling. Figure 7 illustrates the parallel code with 8 cores using static scheduling.

```cpp
#pragma omp parallel for num_threads(8) schedule(static)
for (int i = 0; i < dictionary.size(); i++) {
    if (target == dictionary[i]){
        cout<<"password found: "<< dictionary[i]<<endl;
        cout<<"number of i is: "<< i<<endl;
        cout<<"Number of thread is: "<< omp_get_thread_num()<<endl;
        cout<<"Total number of passwords is: "<<dictionary.size()<<endl;
    }
}
```

**Figure 7.** A segment of parallel code with 8 cores using static scheduling.

4.  Parallel code with 8 cores using dynamic scheduling in one chunk. Figure 8 demonstrates the parallel code with 8 cores using dynamic scheduling in one chunk.

```cpp
#pragma omp parallel for num_threads(8) schedule(dynamic)
for (int i = 0; i < dictionary.size(); i++) {
    if (target == dictionary[i]){
        cout<<"password found: "<< dictionary[i]<<endl;
        cout<<"number of i is: "<< i<<endl;
        cout<<"Number of thread is: "<< omp_get_thread_num()<<endl;
        cout<<"Total number of passwords is: "<<dictionary.size()<<endl;
    }
}
```

**Figure 8.** A segment of parallel code with 8 cores using dynamic scheduling in one chunk.

5.  Parallel code with 8 cores using dynamic scheduling in two chunks. Figure 9 illustrates the parallel code with 8 cores using dynamic scheduling in two chunks.

```cpp
#pragma omp parallel for num_threads(8) schedule(dynamic,2)
for (int i = 0; i < dictionary.size(); i++) {
    if (target == dictionary[i]){
        cout<<"password found: "<< dictionary[i]<<endl;
        cout<<"number of i is: "<< i<<endl;
        cout<<"Number of thread is: "<< omp_get_thread_num()<<endl;
        cout<<"Total number of passwords is: "<<dictionary.size()<<endl;
    }
}
```

**Figure 9.** A segment of parallel code with 8 cores using dynamic scheduling in two chunks.

6.  Parallel code with 8 cores using dynamic scheduling in four chunks. Figure 10 demonstrates the parallel code with 8 cores using dynamic scheduling in four chunks.

```
#pragma omp parallel for num_threads(8) schedule(dynamic,4)
for (int i = 0; i < dictionary.size(); i++) {
    if (target == dictionary[i]){
        cout<<"password found: "<< dictionary[i]<<endl;
        cout<<"number of i is: "<< i<<endl;
        cout<<"Number of thread is: "<< omp_get_thread_num()<<endl;
        cout<<"Total number of passwords is: "<<dictionary.size()<<endl;
    }
}
```

**Figure 10.** A segment of parallel code with 8 cores using dynamic scheduling in four chunks.

*3.4. Applying Brute Force and Dictionary Attacks Using Hashcat #4*

In this technique, we used the Hashcat tool version 6.2.6 to launch two types of attacks, namely the brute force and dictionary attacks. The purpose of this investigation is to evaluate the influence of different GPUs on password cracking. Specifically, one of the GPUs used in this study supports CUDA, and we compare the performance of the two GPUs, namely Intel(R) HD Graphics 630 and NVIDIA(R) GeForce(R) GTX 1050Ti with 4GB GDDR5 and 768 CUDA cores. In addition, we evaluate the impact of password charset on the speed of password cracking and how password structure affects the process. By employing this approach, we aim to gain insights into the effectiveness of different GPUs and password characteristics in password cracking. The following Table 2 demonstrates the hardware and software specifications.

**Table 2.** Hardware and Software Configurations.

| | | |
|---|---|---|
| Hardware | CPU | 7th Generation Intel(R) Core(TM) i7-7700HQ Quad Core (6 MB Cache, up to 3.8 GHz) |
| | Memory | 16 GB, DDR4, 2400 MHz |
| | Storage | 1 TB 5400 rpm hard drive + 128 GB solid state drive |
| | GPU 0 | Intel(R) HD Graphics 630 |
| | GPU 1 | NVIDIA(R) GeForce(R) GTX 1050 Ti with 4 GB GDDR5 with 768 CUDA cores |
| Software | Operating System | Windows 11 Home 64-bit (10.0, Build 22,000) |
| | Driver | GeForce Driver 531.68 |
| | Software | Hashcat v6.2.6 |

A collection of passwords was generated randomly and secured using the MD5 encryption technique. We utilized the same passwords that were in previous techniques. Each password's string and corresponding hash value are illustrated in Table 3. Subsequently, we attempted to crack the passwords by providing only their hash values to the Hashcat tool.

Hashcat is employed by specifying a script or command line with different parameters based on the experimental objective. The "-m" parameter specifies the encryption type, which is 0 (MD5) in this case. The "-a" parameter determines the type of attack to be conducted, with 3 for brute force attack and 0 for dictionary attack. The "d" parameter selects the device to be utilized for password cracking, where "3" is for CPU + GPU 0, and "1" is for GPU 1 (CUDA) exclusively. Finally, the character set for the experiment is determined using the "l" parameter for lowercase English characters, "u" for uppercase English characters, "d" for digit numbers, and "s" for special characters. We aim to test the impact of including special characters on password-cracking performance. Furthermore, it is our intention to assess the efficacy of various hardware setups for the purpose of executing these attacks. Table 4 shows the scripts used for different cracking scenarios.

**Table 3.** A list of randomly generated passwords.

|  | **Password String** | **Hash Value (MD5)** |
|---|---|---|
| Password 1 | abc14 | b80eaaf275cf1d34b88d0b8c6c7da20b |
| Password 2 | hd180 | decce0ac22fc85a9899a1f8ba2c08bfb |
| Password 3 | a7ro1 | 0fd4d72214cd938a1bef4e1a58f4366f |
| Password 4 | tynq0 | 8ea15dcb8862ccab2fa6388fb43317f6 |
| Password 5 | o3kli | 670244cdc900710194338673e26dba1f |
| Password 6 | v1m5e | bc4a21024aa58f2558a1e98e5839e54d |
| Password 7 | x1z5l | 39db818049350277c4400cb01dd3f112 |
| Password 8 | asdf32 | cb697d6d9fbd75cb15fb4670c5aaf0ca |
| Password 9 | f6d3a1 | 87c818c75041578020d71acd4c2ea79f |
| Password 10 | j3g3v1 | 4be9b447aa3d00aefcc69629b626d460 |

**Table 4.** Brute Force and Dictionary Attack Scripts.

| | | |
|---|---|---|
| **Brute Force Attack** | Charset | abcdefghijklmnopqrstuvwxyzABCDEFGHIJKLMNOPQRSTUVWXYZ0123456789 «space»!"#$%&'()*+,-./:;<=>?@[\]^_'{ǀ}~ |
| | Script for CPU + GPU 0 | hashcat -m 0 -a 3 -d 3 hashedpasswords.txt -o cracked.txt -1 ?l?u?d?s ?1?1?1?1?1?1 |
| | Script with GPU 1 (CUDA) | hashcat -m 0 -a 3 -d 1 hashedpasswords.txt -o cracked.txt -1 ?l?u?d?s ?1?1?1?1?1?1 –increment |
| **Dictionary Attack** | Charset | abcdefghijklmnopqrstuvwxyzABCDEFGHIJKLMNOPQRSTUVWXYZ0123456789 «space»!"#$%&'()*+,-./:;<=>?@[\]^_'{ǀ}~ |
| | Script for CPU + HD Graphics (Intel) | hashcat -m 0 -a 0 -d 1 hashedpasswords.txt rockyou.txt -o cracked.txt |
| | Script with GPU 1 (CUDA) | hashcat -m 0 -a 0 -d 3 hashedpasswords.txt rockyou.txt -o cracked.txt |

## 4. Empirical Studies

### 4.1. Description of the Dataset

The dictionary attack can be performed by having a list of passwords. Therefore, we used a list of 14,442,064 unique passwords from RockYou available on Kaggle [27]. According to Kaggle, in 2009, a gaming company called RockYou suffered a hacking incident. The severity of the breach was magnified due to the company's lack of security measures, as they kept all their passwords in clear text and unencrypted, making them easily accessible to the attacker. The hackers were able to obtain a list of all the passwords and made it publicly available. The RockYou dictionary is reputed to be one of the largest and most recognizable leaked collections of passwords [4].

### 4.2. Experimental Setup

We conducted four experiments in our research. The first experiment involved performing brute force using parallelism in Python. In the second experiment, we wrote code using Python to perform a dictionary attack. Then, the results of these two experiments were compared. However, in the third experiment, we performed a dictionary attack by implementing OpenMP in C++ and modified some parameters in the clauses, such as the number of threads, the scheduling type, and the number of chunks. It is also noticed that the experience of running the code is very similar. The user will be asked to insert the password intended to be tested, and the results will be displayed to them in detail. Each technique was set up differently. Lastly, the fourth experiment was conducted by using

Hashcat to evaluate the performance of the hardware configurations of two different GPUs. The following subsections present the setup process for each technique.

### 4.2.1. Brute Force Setup

For brute force, the experiment was carried out using Python with the use of the "ProcessPoolExecutor" class available in the "concurrent.futures" module. The program first prompts the user to enter the targeted password and the number of CPU cores to use. Then, it creates a list of all possible combinations of letters and numbers of increasing length up to the length of the target password. Each combination is passed to the "attempt_match" function to check if it matches the target. The process of checking each combination is performed parallelly with the help of the "ProcessPoolExecutor" from the "concurrent.futures" library, which allows for parallel execution of tasks by distributing them across the specified number of CPU cores. The function returns the first combination that matches the target password, and the total time taken to complete the process is recorded and displayed at the end. The parameters used are as follows:

- "length": This is a parameter for the "attempt_match" function that specifies the length of the password combinations being generated and checked.
- "target": This is a parameter for the attempt_match function that specifies the target password to be cracked. It can either be passed as an argument or obtained by user input.
- "Numb_of_cores": This is a parameter obtained by user input that specifies the number of CPU cores to be used for the concurrent execution of the password-checking task. It is passed to the ProcessPoolExecutor class as the max_workers argument.
- "start_time": This is a variable that stores the current time when the password-cracking process starts.
- "end_time": This is a variable that stores the current time when the password-cracking process ends.
- "total_time": This is a variable that stores the time elapsed between start_time and end_time and represents the total time taken to complete the password-cracking process.

### 4.2.2. Dictionary Attack Setup

The dictionary attack consists of different experiments. The first one was implemented with Python to compare it with brute force (first experiment). The second one was performed with the use of OpenMP to see the different performances acquired when changing the parameters of the clauses.

For the first experiment, we used Python to set up a parallelized password-cracking experiment. The experiment uses the Python standard library's "multiprocessing" module to create and manage child processes. The child processes the "password_check" function parallelly, each checking a portion of a dictionary of possible passwords to find a match with the target password. The main program, defined in the "main" function, starts by reading a target password from the user and then loads a dictionary of possible passwords from a file we named "dictionary.txt". The dictionary is split into "num_processes" equal parts (8 parts in this case, and it can be changed), and "num_processes" child processes are started to parallelly check each part of the dictionary. Each child process is passed the dictionary, the target password, and the indices marking the start and end of the portion of the dictionary it should check as arguments to the "password_check" function. If a password match is found, the "password_check" function outputs information about the matching password and the elapsed time, then returns. After starting the child processes, the main program waits for all child processes to finish by calling join on each. The code uses the "time" module to measure the elapsed time of password checking. These are the parameters that were used:

- "dictionary": a list of strings representing a dictionary of possible passwords.
- "target": a string representing the target password to be found.

- "start_index": an integer representing the starting index of the portion of the dictionary to be checked by a child process.
- "end_index": an integer representing the ending index of the portion of the dictionary to be checked by a child process.
- "num_processes": an integer representing the number of child processes to be created and used for parallelizing password checking.
- "chunk_size": an integer representing the size of each chunk of the dictionary, calculated as the length of the dictionary divided by "num_processes".
- "processes": a list to store the created child processes.
- "file": a file object representing the "dictionary.txt" file.
- "passwd": a string representing a password from the "dictionary.txt" file, read one at a time in a loop.
- "p": a process object representing a child process.

For the second experiment, we wrote a code that functions similarly to the Python code. However, the purpose here is to compare different settings of using OpenMP, such as the number of cores and the type of schedule. The program reads a list of potential passwords from a file named "dictionary.txt" (RockYou passwords) and stores it in a vector called "dictionary". The user is prompted to enter the target password, which is stored in the variable "target". The program then uses the "#pragma omp parallel for" directive to parallelize a for-loop that iterates through the elements of the "dictionary" vector and compares each one to the target password. The "num_threads" clause sets the number of threads to any number you wish to test depending on your machine, and the "schedule(Typeofschedule, NumberOfChuncks)" sets the scheduling method for distributing iterations among the threads to be the type of schedule you are willing to test (static or dynamic) with a chunk size that you can set. If a match is found, the program outputs the matching password and the number of the iteration in the for-loop that found the match, as well as the number of threads that found the match and the total number of passwords in the dictionary. Finally, the program measures and outputs the elapsed time in milliseconds using the "chrono" library. The following are the parameters used:

- "num_threads(8)": This sets the number of threads to be used in the parallel for-loop. The value of 8 means that 8 threads will be created to execute the loop in parallel.
- "schedule(type, chunk)": This sets the scheduling method for distributing iterations among the threads with the number of chunks.
- "target": This is the target password that the program is trying to find in the "dictionary.txt" file. The user is prompted to enter the target password.
- "dictionary": This is a vector that stores the list of potential passwords read from the "dictionary.txt" file.
- "pass": This is a string variable that is used to temporarily store each password read from the "dictionary.txt" file.
- "i": This is the loop variable for the for-loop that iterates through the "dictionary" vector.
- "omp_get_thread_num()": This is a function from the OpenMP library that returns the number of threads executing the code.

## 5. Results and Discussion

The first technique and the second technique were tested on a computer with the following specifications: Intel(R) Core (TM) i5-9400F CPU, a RAM of 16 GB, and NVIDIA GeForce RTX 2060 Graphic Card. The code in both techniques was written in Python. A list of ten passwords with a max length of six characters as a string was generated randomly with small English characters and numbers. The passwords toward the end of the list are more complicated compared to those at the start. The reason for the increased difficulty in cracking these passwords is that they either contain digits (those with digits at the beginning of the string being particularly difficult) or contain trailing English characters. The organization or arrangement of characters within a character set has a significant impact on the ability to crack passwords. A well-structured character set, such as the one

that places English alphabet characters at the beginning followed by digits towards the end, makes it more challenging for an attacker to use brute force techniques to guess a password that begins with trailing characters of the character set. While implementing the brute force technique on very short passwords in a sequential code, we obtained a better score compared with the parallel code. For instance, the first password (the simplest one) took 0.2471 s in serial code and 0.3286 in parallel. However, the longer and more complicated the password becomes, the better the parallel code performs compared to the sequential one. The best improvement obtained was when using the parallel code with the maximum number of cores of the device (six cores). The speedup was 1.9× times. Below, Figure 11 illustrates the implementation of the first technique.

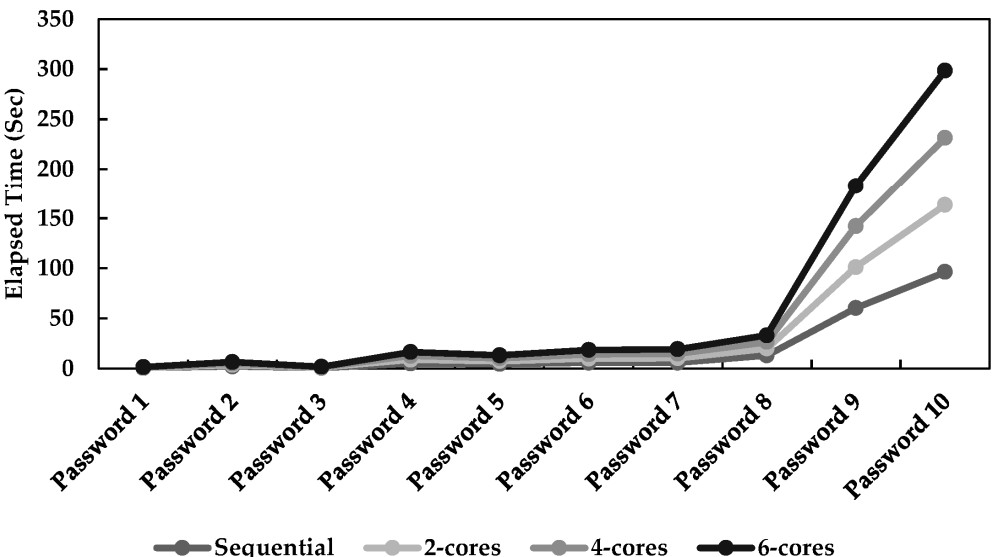

**Figure 11.** Ten various passwords being tested for the Brute Force Attack.

On the other hand, in the implementation of the second technique, "dictionary attack", we decided to test the same list of passwords used with the brute force technique to compare them both. The total number of passwords in the dictionary is almost 14 million, as mentioned previously. Because the passwords tested were generated randomly (are not commonly used), only two passwords were able to be cracked in the dictionary attack technique. There was no improvement noticed between the sequential and parallel codes in Python because of the process performed by the library "ProcessPoolExecutor". The process involves breaking down the task (cracking passwords) and distributing it across multiple cores, takes some time and resources, and may result in additional work or overhead, ultimately canceling out any potential benefits of parallel processing. Nevertheless, the execution time of the dictionary attack was tremendously better than the brute force by almost 930 times. Figure 12 below shows the results of the dictionary attack experiment.

It is clear that the dictionary attack outperforms the brute force attack in terms of time due to the limited number of iterations available (almost 14 million iterations in this case). However, the brute force attack was able to capture all targeted passwords. The length of the passwords in the brute force attack greatly affects the execution time, whereas, in the dictionary attack, a very long common password can be easily cracked. The first password dictionary attack technique with six cores is faster than the brute force attack by approximately 4.79× times. The eighth password was cracked approximately 930× times faster because it was complicated by the brute force technique.

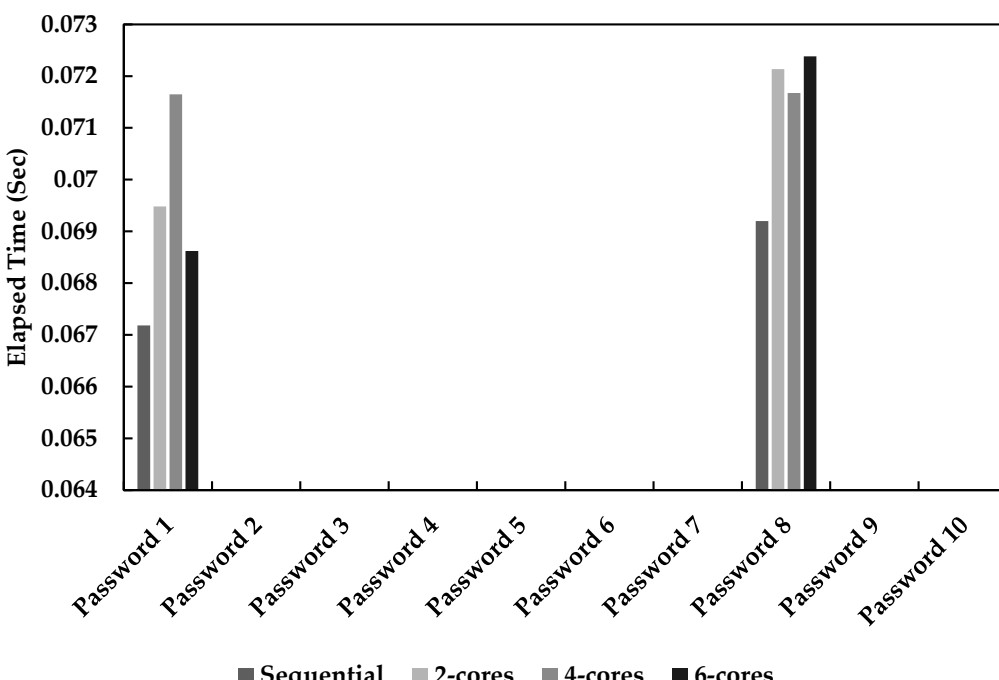

**Figure 12.** Ten various passwords being tested for the Dictionary Attack.

In the third technique, the experiment was conducted on a computer with the following specifications: 11th Gen Intel(R) Core (TM) i5-1135G7, a RAM of 8 GB, and Intel(R) Iris(R) Xe Graphics. The experimental outcome showed that the best parameters of parallel processing in terms of execution time occurred with eight-core static scheduling. The execution time was improved by 4.4× times, going from 181 to 41 in time elapsed. However, the lower the number of chunks used in dynamic scheduling, the slower the execution time becomes compared to the sequential code. In other words, the more chunks we have in the dynamic scheduling, the higher speedup we obtain in comparison with the sequential program. Below is a graph that shows the different settings used on ten different passwords with their elapsed times recorded. Figure 13 below shows the results of the third technique.

In addition, we made sure the output contained the iteration number in which the password was found (equal to the index of the password in the dictionary list), the thread number that processed the list of the entered password, and, most importantly, the elapsed time taken when running the thread. In the screenshot below, one of the tests was captured related to testing the dynamic scheduling with four chunks in eight threads. For the four chunks, the program divides the password list into four equal parts, and each part is processed by a separate thread. This means that the workload is evenly distributed across four threads, which can lead to faster execution times than sequential processing. However, if one thread finishes its part of the workload earlier than the others, it does not have to wait until the remaining threads finish processing their parts because it is dynamic, in contrast to static, where it has to wait for the remaining threads to finish their parts.

In our fourth approach, we aim to examine the efficiency of various GPU types for password cracking using the widely known tool Hashcat, as shown in Figure 14. The results of our study indicate that without including special characters in the Hashcat parameters, GPU 0 (Intel(R) HD Graphics 630) took 23 s to crack the password list mentioned earlier with a speed of 254.8 MH/s, whereas GPU 1 (NVIDIA GeForce GTX 1050 Ti) completed the same task in just 2 s with a speed of 2558.3 MH/s. This means that GPU 1 is approximately 11.5 times faster than GPU 0 for cracking passwords at hand, despite the fact that the machine's CPU supported GPU 0. Regarding the inclusion of special characters in the character set, the combination of CPU and GPU 0 took 197 s to crack the specified password list, while GPU 1 was able to complete the same task in just 19 s. Therefore, GPU 1 is approximately 10.4 times faster than the combination of CPU and GPU 0 for the given

task of cracking the password list. Our results reveal that the elapsed time for cracking passwords increased significantly when special characters were added to the Hashcat parameters. Specifically, the combination of CPU and GPU 0 required 197 s to complete the task, which is significantly longer than the 23 s it took to crack the same password list without special characters. This indicates that the inclusion of special characters in passwords can make the cracking process more challenging. Similarly, GPU1 required 19 s to crack the password list when special characters were added, compared to just 2 s when no special characters were included. Table 5 below shows the brute force attack test results.

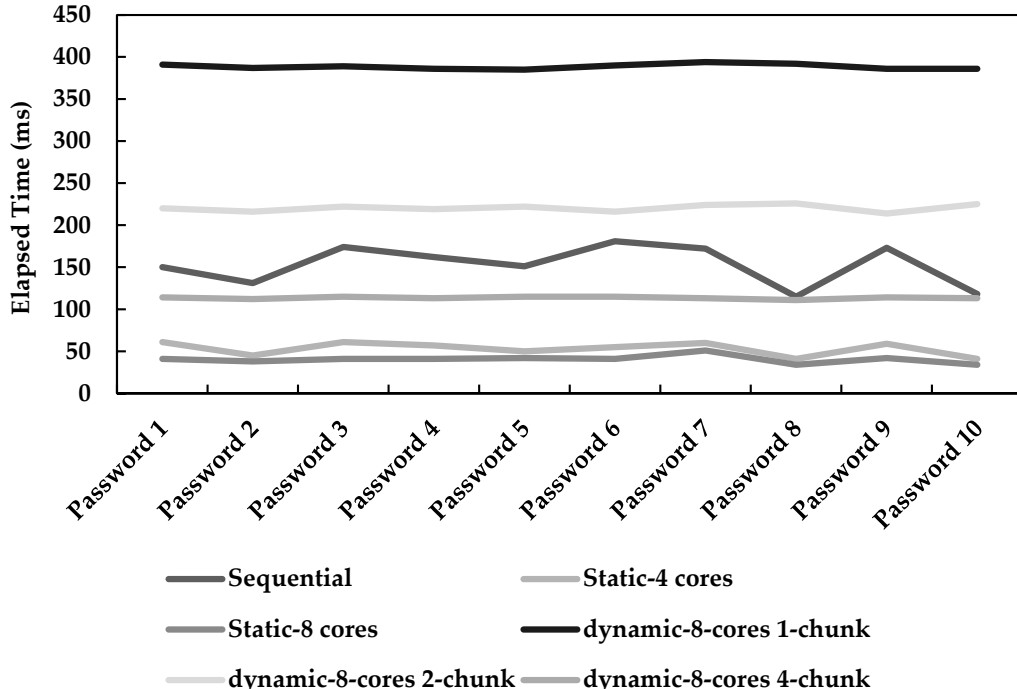

**Figure 13.** A comparison dictionary attack by varying parameters using OpenMP.

**Table 5.** Brute force attack test results.

| Performance Metrics | CPU + GPU 0 (No Spec Chars) | CPU + GPU 0 (with Spec Chars) | GPU 1 Only CUDA (No Spec Chars) | GPU 1 Only CUDA (with Spec Chars) |
|---|---|---|---|---|
| Time (s) | 23 | 197 | 2 | 19 |
| Speed (MH/s) | 254.8 | 244.3 | 2588.3 | 2639.8 |

In the dictionary attack, our experiment demonstrates that GPU 1 (NVIDIA GeForce GTX 1050 Ti) with CUDA performs significantly better than CPU + GPU 0 (the combination of Intel(R) Core(TM) i7-7700HQ CPU and Intel(R) HD Graphics 630). In terms of units, GPU 1 (CUDA) has an average speed of 10,650 kH/s, while the average speed of CPU + GPU 0 is 2864.3 kH/s. The speed difference between the two is substantial. Regarding the elapsed time, CPU + GPU 0 required five more seconds to complete the task, which is three seconds slower than GPU 1 (CUDA). Table 6 shows the dictionary attack test results.

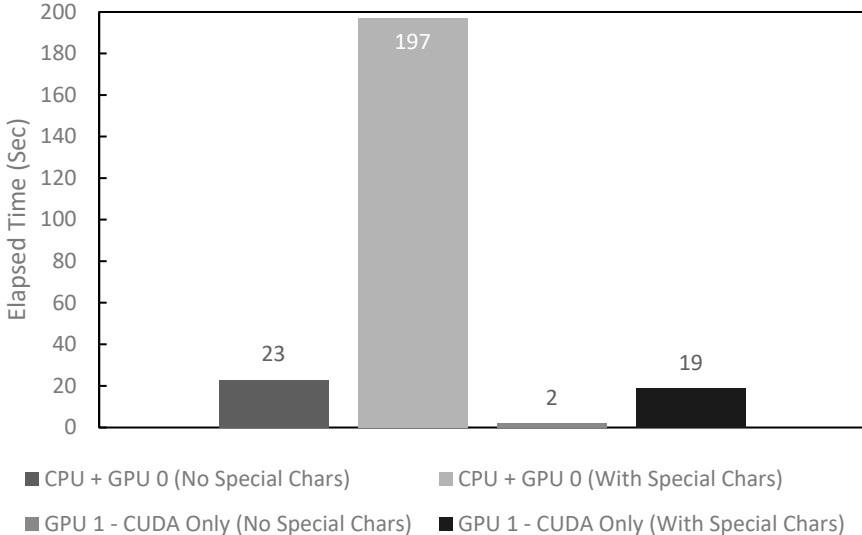

**Figure 14.** A comparison of elapsed time between various hardware configurations and charsets.

**Table 6.** Dictionary attack test results.

| Performance Metrics | CPU + GPU 0 | GPU 1 CUDA Only |
|:---:|:---:|:---:|
| Time (s) | 5 | 2 |
| Speed (kH/s) | 2864.3 | 10,650 |

Table 7 presents improved performance obtained by modifying related parameters for each of the password-cracking techniques mentioned in this work.

**Table 7.** Improved performance obtained with modified parameters.

| | Brute Force Technique with Python #1 | Dictionary Attack with Python Technique #2 | Dictionary Attack with OpenMP in C++ Technique #3 | Brute Force Attack with Hashcat Using GPU 1 (CUDA) #4 |
|:---:|:---:|:---:|:---:|:---:|
| Execution Time | 6.6342 s. | 0.0723827 s. | 0.041 s. | 2 s. |
| Speedup | 1.9× times. | No improvement because of potential overhead caused by the Python library. However, tremendous improvement between the dictionary attack and the brute force attack by almost 930 times. | 4.4× times. | 11.5× times. |

## 6. Comparison and Analysis

Experiments were performed by the authors of [7] using both brute force and dictionary attack methods. The brute force method effectively cracked relatively short passwords (six to seven characters) on SHA-1 password hash. However, the tests demonstrated that brute force was not very successful in cracking passwords consisting of eight or more characters. Similarly, our experiment showed that the longer the passwords are, the longer it takes to crack them successfully by the brute force attack. The authors of [7] implemented their work using a cybersecurity software called "Hashcat" with the 5.1.0 version. Nevertheless, we have written codes in Python and C++ from scratch to perform brute force and dictionary attacks. Moreover, we assessed hardware configurations for the two cybersecurity attacks mentioned using Hashcat version 6.2.6. The password encryption employed was MD5, which differed from the SHA-1 encryption used in [7]. Although the authors of [7]

used a different method of execution and setup to implement their tests, our experiment and their experiment both demonstrated that when attempting to crack notably lengthy passwords, the dictionary attack was more successful than a brute force attack. In the brute force attack, ref. [7] used a character set that only included small and capital English letters, while we augmented this set by adding special characters.

When comparing our work with previous work in [28], it can be seen that both papers aim to explore techniques for parallelizing password cracking, specifically the dictionary attack. Still, there are significant differences between the two in terms of the approach and methodology. Reference [28] focuses on parallelizing the MD5 and SHA1 hashing algorithms using the KASTL library on three different systems with varying hardware specifications. The authors experimented with various word lists of different sizes and added arbitrary words to them. They compared the performance of CPU and GPU processing using different tools and libraries and concluded that GPU processing is much faster than CPU processing. They also had to modify existing code to allow for parallelization using KASTL. On the other hand, our paper compares four different techniques for conducting brute force and dictionary attacks on a list of passwords obtained from the RockYou data breach. The techniques used were ProcessPoolExecutor in Python, multiprocessing module in Python, and OpenMP in C++. We compared sequential and parallel execution times for each technique and concluded that multiprocessing in Python was the most efficient for the dictionary attack, while OpenMP in C++ was the most efficient for brute force attacks.

## 7. Conclusions

In conclusion, the results of these four experiments revealed that the performance of password-cracking techniques could be significantly enhanced by utilizing parallel processing techniques on hardware configurations such as multiple cores or powerful GPUs. The efficiency of GPU types was examined using Hashcat. NVIDIA GeForce GTX 1050 Ti was found to be approximately 11.5 times faster than Intel(R) HD Graphics 630 in cracking passwords, and when special characters were added to the character set, the former was approximately 10.4 times faster than the combination of CPU and GPU 0. In terms of the dictionary attack, GPU 1 (CUDA) performed significantly better than CPU + GPU 0, with an average speed of 10,650 kH/s. On the other hand, one experiment showed that the parallel code performed better compared to the sequential code, and the speedup was $1.9\times$ times with six cores. Another experiment showed that the best performance in terms of the execution time was achieved with eight-core static scheduling, with a speedup of $4.4\times$ times. In addition, dynamic scheduling with a higher number of chunks produces a better performance compared to sequential code. These findings emphasize the importance of using advanced hardware and parallel processing techniques to improve the efficiency of password cracking. As cyberattacks become increasingly sophisticated, it is essential that security professionals have access to the most powerful and efficient tools possible. The use of GPUs and parallel processing techniques can greatly enhance the speed and accuracy of password cracking, which is a critical component of many cyberattacks.

**Author Contributions:** Writing—original draft, I.A., M.A. (Mohammed Albugami), A.A. (Ali Alkhwaja), M.A. (Mohammed Alghamdi), H.A. and F.A.; writing—review and editing, A.A. (Abdullah Almurayh) and N.M.-A. All authors have read and agreed to the published version of the manuscript.

**Funding:** This research received no external funding.

**Institutional Review Board Statement:** Not applicable.

**Informed Consent Statement:** Not applicable.

**Data Availability Statement:** Not applicable.

**Acknowledgments:** The authors wish to acknowledge the anonymous reviewers for providing valuable feedback on the initial versions of the manuscript. The authors would like to acknowledge the Department of Computer Science, College of Computer Science and Information Technology, Imam Abdulrahman Bin Faisal University, for supporting and facilitating this research.

**Conflicts of Interest:** The authors declare no conflict of interest.

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
