# Peer review of "Password Cracking with Brute Force Algorithm and Dictionary Attack Using Parallel Programming"

_applsci, doi:10.3390/app13105979_

Round 1

Reviewer 1 Report (Previous Reviewer 3)

Dear authors

There has been quite a number of improvement in the revised submission 

It adds clarity to the paper.

Reviewer 2 Report (Previous Reviewer 2)

All the previously mentioned comments were considered and the manuscript has been changed. No additional changes are needed.

This manuscript is a resubmission of an earlier submission. The following is a list of the peer review reports and author responses from that submission.

Round 1

Reviewer 1 Report

Comment 1:

"In conclusion, the results of these three experiments showed that parallel processing could greatly improve the speed of password-cracking techniques" - what is novel about this? Seems obvious! 

"The second experiment demonstrated that the dictionary attack was faster than the brute force attack" - what is novel about this? Seems obvious! 

Comment 2:

Why is clear text password cracking being studied? The dictionary attacks and brute force attacks should target salted/hashed passwords to be relevant.

Comment 3:

GPUs are used by prior work - why doesn't this work consider it?

Comment 4:

Why is std::thread not used with C++? Why is there no speed up for dictionary attacks with parallelization with Python? No analysis is presented. The results are not convincing. 

Reviewer 2 Report

The introduction part is missing with the main research problem, which the authors are trying to solve in this paper. The novel contributions need to be described here as well. It is not clear what novel solution the authors are presenting in this paper. The implementation of the well-known brute-force and dictionary attacks using different programming languages is not sufficient to be published in a scientific journal.
It is not clear why the authors compare their results with the results of a work that was published quite long ago (14 years ago). I strongly recommend finding the newest reference in this research field and compare the achieved results with it.
Figure 1, 3 are in poor quality, it needs corrections.
The reference to Kaggle should be added in Section 4.1. 
The scenarios for the performed experiments are not clearly described. Moreover, the figures provide the results for sequential, static-4, static-8, etc. techniques. The details of how these techniques are implemented should be described. What 10 passwords were used? How was the attack performed for each of the two techniques? How has the structure of the password impacted the speed of its cracking? This part should be described more thoroughly.
"There was no improvement between the sequential and parallel codes because the list’s size was small." "Nevertheless, the execution time was tremendously better than the Brute Force." What means small or better? Can the authors provide more clear units?
It is not clear what "optimal parameters mean (see Table 2)? How was optimization was done?
Table 2 provides execution time. It is not clear whether it is the average unit of time, or..? The same goes for the speedup.
The discussion part should be provided in the paper.

Reviewer 3 Report

Dear Authors

The study presented here describes the use of python and parallel programming as part of this study. The study is interesting. However the quality of this work needs to improve substantially in the core area of the research. The authors does provide and explain results section. 
